# Gut Microbiota in Hypertension and Atherosclerosis: A Review

**DOI:** 10.3390/nu12102982

**Published:** 2020-09-29

**Authors:** Barbara J. H. Verhaar, Andrei Prodan, Max Nieuwdorp, Majon Muller

**Affiliations:** 1Department of Internal Medicine, Section Geriatrics, Amsterdam Cardiovascular Sciences, Vrije Universiteit Amsterdam, Amsterdam UMC, 1011-1109 Amsterdam, The Netherlands; majon.muller@amsterdamumc.nl; 2Department of Internal Medicine, Section Vascular Medicine, Universiteit van Amsterdam, Amsterdam UMC, 1011-1109 Amsterdam, The Netherlands; a.prodan@amsterdamumc.nl (A.P.); m.nieuwdorp@amsterdamumc.nl (M.N.)

**Keywords:** gut microbiota, cardiovascular disease, hypertension, atherosclerosis

## Abstract

Gut microbiota and its metabolites such as short chain fatty acids (SCFA), lipopolysaccharides (LPS), and trimethylamine-N-oxide (TMAO) impact cardiovascular health. In this review, we discuss how gut microbiota and gut metabolites can affect hypertension and atherosclerosis. Hypertensive patients were shown to have lower alpha diversity, lower abundance of SCFA-producing microbiota, and higher abundance of gram-negative bacteria, which are a source of LPS. Animal studies point towards a direct role for SCFAs in blood pressure regulation and show that LPS has pro-inflammatory effects. Translocation of LPS into the systemic circulation is a consequence of increased gut permeability. Atherosclerosis, a multifactorial disease, is influenced by the gut microbiota through multiple pathways. Many studies have focused on the pro-atherogenic role of TMAO, however, it is not clear if this is a causal factor. In addition, gut microbiota play a key role in bile acid metabolism and some interventions targeting bile acid receptors tend to decrease atherosclerosis. Concluding, gut microbiota affect hypertension and atherosclerosis through many pathways, providing a wide range of potential therapeutic targets. Challenges ahead include translation of findings and mechanisms to humans and development of therapeutic interventions that target cardiovascular risk by modulation of gut microbes and metabolites.

## 1. Introduction

Cardiovascular diseases, including atherosclerosis and hypertension, are public health care priorities of the World Health Organization (WHO) [1]. Cardiovascular disease is the leading cause of mortality, representing a third of global deaths, and disproportionally affects low- and middle-income countries [2]. Despite current preventive and therapeutic strategies, mortality due to cardiovascular disease is expected to further increase over the next decade [2]. Accumulating evidence describes the role of gut microbiota in cardiovascular disease, potentially providing novel therapeutic targets. The gut microbiome consists of more than 100 trillion micro-organisms, predominantly bacteria and viruses [3]. Due to the development of 16S rRNA gene amplicon sequencing and shotgun metagenomic sequencing, the understanding of the role of the gut microbiota in health and disease has increased tremendously over the past decade [4]. Gut microbiota composition is largely determined by exposure to dietary factors, but conversely, gut microbiota are needed for digestion of macronutrients and production of a wide range of metabolites [5]. Alterations in gut microbiota composition have been observed in a variety of health conditions, including type 2 diabetes, inflammatory bowel disease, asthma, psychiatric disorders, but also in cardiovascular disease [6,7,8,9,10]. In addition, several gut metabolites have been shown to interact with metabolism and the nervous system, affecting insulin sensitivity, energy balance, and appetite regulation [11,12,13].

Low-grade chronic inflammation contributes to the development of both atherosclerosis and hypertension [14,15,16,17]. Gut microbiota can induce systemic inflammation, as has been shown in patients with type 2 diabetes [18]. In addition, gut microbiota could affect cardiovascular risk indirectly, through metabolites such as short chain fatty acids (SCFA) and trimethylamine N-oxide (TMAO). The relation between gut microbiota and its key metabolites in hypertension and atherosclerosis could improve our understanding of differences in susceptibility for cardiovascular disease and provide potential therapeutic targets. In this narrative review, we will focus on the role of gut microbiota in hypertension and atherosclerosis. After summarizing the current evidence, we will discuss future perspectives in this field.

## 2. Gut Microbiota in Hypertension

### 2.1. Gut Microbiota Composition in Hypertension

Hypertension is the most important modifiable risk factor for cardiovascular disease [19]. Although hypertension is thought to be driven by a combination of genetic and lifestyle factors, genome-wide association studies showed that only a small (<5%) proportion of the incidence of hypertension can be explained by genetics [20]. In contrast, lifestyle tends to have a much larger influence, with separate lifestyle factors such as body mass index (BMI) and salt intake affecting blood pressure levels with 5 mmHg [21]. Several dietary interventions, including diets such as the Mediterranean diet and the DASH (Dietary Approaches to Stop Hypertension) diet have illustrated that higher intake of fruits, vegetables, and fibers are associated with lower blood pressure [22,23]. The Mediterranean diet has been shown to induce a rise in SCFAs, key metabolites produced by the gut microbiome [24].

Several animal studies have reported compositional differences in the gut microbiota of animal models for hypertension, including Dahl-sensitive rats, spontaneous hypertensive rats, angiotensin-II induced hypertensive rats, and deoxycorticosterone acetate (DOCA)-salt mice, when compared to wild-type animals [25,26,27,28]. These differences include a lower abundance of SCFA-producing bacteria, higher abundance of lactate-producing bacteria [27], lower abundance of Bacteroidetes, and higher abundance of Proteobacteria and Cyanobacteria [28] compared to control animals. Intervention studies in animals showed that blood pressure levels in these animal models for hypertension can be modified by fecal microbiota transplants and antibiotic treatment [27].

In humans, several cross-sectional studies have assessed associations between gut microbiota composition and blood pressure or hypertension (Table 1) [27,29,30,31,32,33,34,35,36,37]. Despite differences in sequencing methods and downstream analyses, some results regarding microbial alpha diversity and microbiota composition are consistent across studies. Higher blood pressure was associated with lower gut microbiota alpha diversity in almost all studies [27,30,32,34,35,36,37]. Low alpha diversity is considered an adverse but nonspecific characteristic, since a decrease in diversity has also been observed in obesity, hyperinsulinemia, and dyslipidemia. In addition, higher abundances of Gram-negative microbiota including *Klebsiella*, *Parabacteroides*, *Desulfovibrio*, and *Prevotella* were associated with higher blood pressure. Gram-negative bacteria are a source of lipopolysaccharides (LPS), also known as endotoxins, that are pro-inflammatory. In contrast, SCFA-producing bacteria, including *Ruminococcaceae*, *Roseburia*, and *Faecalibacterium* spp. were less abundant in hypertensive compared to normotensive patients [29,31,34,35,37]. Of note, the majority of these studies did not adjust for important confounders such as age, BMI, or dietary factors in their analyses.

Dietary salt intake affects both the incidence of hypertension as well as gut microbiota composition. Higher salt intake has been associated with a shift in microbiota composition in several animal models, including an increase in *Lachnospiraceae*, *Ruminococcus*, and *Parasutterella* spp. and decrease in *Lactobacillus* and *Oscillibacter* [38,39,40]. *Lactobacillus* abundance has been associated with salt sensitivity in hypertension, since supplementation of *Lactobacillus* spp. in a mice model has been shown to attenuate salt-sensitive hypertension, presumably by modulation of Th17-cells [40]. The blood pressure lowering effect of *Lactobacillus* was confirmed by several other animal models [41,42,43,44]. In humans, however, a decrease of *Lactobacillus* spp. was only reported by one of the cross-sectional studies in hypertensive subjects in Table 1 [29]. A meta-analysis including nine randomized-controlled trials, predominantly with healthy controls, found a blood pressure lowering effect of probiotics with several *Lactobacillus* spp. [43]. The blood pressure lowering effect tended to be stronger in the only included placebo-controlled intervention study with hypertensive subjects (17/13), although this study did not assess changes in gut microbiota composition [45].

In summary, animal studies suggest a causal link between gut microbiota composition and blood pressure regulation. Cross-sectional studies in human subjects show specific differences in microbiota composition between hypertensive subjects and controls, including lower SCFA-producing bacteria and higher Gram-negative species. These differences point to a role for SCFAs and LPS in hypertension, although the direction of this association is unclear.

### 2.2. Short Chain Fatty Acids

SCFAs, including acetate, propionate, and butyrate, are produced by specific gut microbes by fermentation of otherwise indigestible dietary fibers [46]. Fecal and plasma levels of SCFA are associated with the abundance of SCFA-producing microbiota in the gut and the intake of dietary fibers [36,47,48]. Butyrate-producing microbiota include bacteria from the families *Ruminococcaceae* and *Lachnospiraceae*, but also bacteria such as *Anaerobutyricum hallii* and *Anaerostipes* spp. Acetate and propionate are mainly produced by *Bifidobacterium* spp. and mucin-degrading bacteria such as *Akkermansia muciniphila* [49]. Most of the produced acetate and propionate is absorbed by the gut, while butyrate is used as a primary energy source by colonocytes and only absorbed in very small proportions [50,51]. As a result, plasma concentrations of acetate and propionate are much higher than circulating butyrate levels.

Human studies on the role of SCFAs in blood pressure regulation are rather scarce. Intriguingly, fecal SCFA concentrations in humans have been associated with higher blood pressure [30], while SCFA-producing microbiota are often associated with lower blood pressure [31,35,37]. Perhaps, increased SCFA availability in the intestines results in upregulation of absorption mechanisms, which could lead to relatively lower fecal concentrations and higher plasma availability, as was supported by a murine model [52]. There are no results from human intervention studies with SCFAs to target blood pressure. However, butyrate tended to lower blood pressure in intervention trials in subjects with metabolic syndrome [53,54]. Moreover, the Mediterranean diet, which induces a rise in SCFA levels, has been reported to have a blood pressure lowering effect [24].

In animal models, SCFAs were associated with both higher and lower blood pressure, which might be explained by the differential effects of SCFA receptors [55]. Several SCFA receptors have been identified, including fatty acid receptor (FFAR)-2 and FFAR3 (formerly known as GPR43 and GPR41) [56]. Animal studies have shown that SCFAs can have differential effects on blood pressure depending on the receptors involved. FFAR2 is expressed in a variety of tissues, including renal arteries, and causes vasodilation in response to SCFAs. In contrast, a blood pressure elevating effect is mediated by Olfr78 in mice through renin release from granules in the renal juxtaglomerular apparatus [57,58]. The potency of SCFAs is much lower for Olfr78 and the human analogue, OR51E2, than for FFAR2, and therefore, it was suggested that Olfr78 serves as a negative feedback loop for the blood pressure lowering effects of FFAR2 [59].

In addition, SCFAs, in particular butyrate, have anti-inflammatory effects that are presumed to be mediated by inhibition of histone deacetylase (HDAC) [60,61]. Butyrate suppresses the production of pro-inflammatory cytokines, such as tumor-necrosis factor-α (TNF-α), interleukin-12 (IL-12), and interferon-γ (IF-γ), and upregulates the production of anti-inflammatory interleukin-10 (IL-10) by monocytes in vitro [62]. In addition, SCFAs have anti-inflammatory effects on epithelial cells that are partly mediated through HDAC [63]. In spontaneously hypertensive rats, HDAC activation has been associated with hypertension [64]. Conversely, butyrate administration to mice resulted in decreased blood pressure levels and reduced renal inflammation by HDAC inhibition [65].

SCFAs have also been suggested to be implicated in gut–brain communication. Vagal afferents express receptors that can sense SCFAs, which provides another pathway for the blood pressure modulating effects of SCFAs [66]. Animal studies showed that higher colonic levels of acetate could result in blood pressure lowering through parasympathetic activation. In addition, the blood pressure lowering effects of butyrate in rats were shown to be significantly reduced by vagotomy [67]. Another study with spontaneous hypertensive rats described a reduced central responsiveness to butyrate, as a result from reduced expression of butyrate receptors in the hypothalamus [52]. Thus, SCFAs could affect blood pressure through direct vascular and renal receptors, through HDAC inhibition, but also through colonic nerve signaling.

### 2.3. Gut Permeability and Lipopolysaccharides

Gut microbiota can also affect gut permeability and therefore influence the extent to which metabolites and endotoxins are absorbed (Figure 1). The barrier of the intestinal epithelium consists primarily of enterocyte brush borders and is more permeable for hydrophobic than for water soluble compounds. However, intercellular junctions on the enterocyte’s lateral margins provide an alternative paracellular absorption route [68]. These intercellular junctions are dynamic structures that regulate paracellular permeability, and consist of tight junctions on the luminal side and adherens junctions on the laminal side. The level of permeability can be influenced by dietary factors, but also by the zonulin pathway. Zonulin is secreted by the basal lamina of the intestinal epithelium and binds enterocytes to initiate a complex intracellular signaling pathway that eventually phosphorylates the tight junction, resulting in permeability of the paracellular route [69]. Gut microbiota such as *Vibrio cholerae* appear to exploit this physiological pathway by excreting zona occludens toxin, a zonulin homologue that has similar effects [70].

Animal models suggest that gut permeability is higher in the hypertensive state. Hypertensive rats had lower levels of mRNA of gap junction proteins, indicating higher gut permeability, which was restored after fecal microbiota transplantation from controls [71]. In a similar model, an increase in blood pressure in spontaneous hypertensive rats was associated with more permeability and lower levels of tight junction proteins [72].

A consequence of higher gut permeability is increased translocation of certain metabolites and endotoxins in the portal and systemic circulation, which could cause further amplification of gut permeability [73]. Lipopolysaccharides (LPS), also known as endotoxins, can be found in the outer membrane of Gram-negative bacteria, the most abundant bacteria in the gut microbiome [74]. The lipid A component of LPS is the main pathogen-associated molecular pattern (PAMP) that can interact with Toll-like receptor 4 (TLR4) [75,76]. When translocated from the gut into the circulation, LPS forms a complex with LPS-binding protein (LBP) which can bind to CD14 on mononuclear cells [77]. This could lead to production of pro-inflammatory cytokines, such as TNF-α, interleukin-1 (IL-1), and interleukin-6 (IL-6), mediated by the MD2/TLR4 receptor complex [76,78]. Butyrate was shown to attenuate the pro-inflammatory effects of LPS-stimulation [79].

LPS is known to induce systemic inflammation and has been shown to have both metabolic and cardiovascular effects. In mice, infusion of LPS to 2- to 3-fold higher plasma levels resulted in higher glucose and insulin levels and weight gain comparable to mice on a 4-week high-fat diet [73]. LPS-administration to rats increased heart rate and norepinephrine levels, decreased baroreflex sensitivity, and increased neuroinflammation, as indicated by increased TLR and TNF-alfa expression in the paraventricular nucleus (PVN) that plays a key role in blood pressure regulation [80]. The same effects were observed in a small (*n* = 8) group of human subjects that showed a significant decrease in systolic and diastolic blood pressure after administration of LPS. Moreover, in this study, LPS increased brain microglial activation on positron emission tomography (PET)-scans [81]. Summarizing, there is a limited number of studies suggesting that systemic LPS could have pro-inflammatory, sympathetic activating, and neuroinflammatory effects, all of which are relevant in hypertension pathogenesis.

### 2.4. Gut-Brain Interactions and Sympathetic Activation

Increased sympathetic activation is considered one of the causal factors in the development of hypertension, and can already be observed in early stages [82]. The sympathetic nervous system modulates blood pressure levels through vasoconstriction in peripheral blood vessels, renal regulation of water and sodium balance, and release of renin by juxtaglomerular cells [83]. Regions in the central nervous system that are involved in sympathetic activation include the PVN, the nucleus of the solitary tract (NTS), and the rostral ventrolateral medulla (RVLM) [84]. Hypertension is associated with neuroinflammation in these regions, which might be mediated by the renin-angiotensin aldosterone system, since prorenin was shown to cause microglial activation in mice and spontaneously hypertensive rats (SHR) [85,86].

Gut–brain communication could stimulate sympathetic activation and therefore play a role in the hypertension pathogenesis. The gut is innervated by the autonomic nervous system that signals physiological conditions such as acidity, osmolarity, and pain [87]. Intrinsically, the enteric nervous system (ENS), consisting of the myenteric plexus and the submucosal plexus, controls intestinal motor and sensory functions [88]. The ENS is a complex system that is sometimes referred to as the ‘second brain’, because of the structural and functional similarities [89]. It communicates with the brain via the vagal nerve, which projects to the NTS, that is involved in sympathetic regulation. Gut microbiota interfere in ENS–brain interactions by stimulating enterochromaffin cells to produce serotonin, a neurotransmitter that affects gut secretion, motility, and local nerve reflexes [90]. Conversely, central sympathetic activation can, through a cascade of events, lead to increased gut permeability and increased translocation of metabolites into the systemic circulation [91].

Elevated sympathetic drive shifts bone marrow hemopoietic stem cells to a pro-inflammatory state, and the release of these immune cells contributes to further hypertension development [92,93]. An animal study with SHR showed that microbiota affect inflammation in brain regions crucial to sympathetic outflow. Microbiota composition in these rats was associated with reactive oxygen species (ROS) and proinflammatory cytokines in the PVN [71]. In addition, fecal transplantation in rat models from Wistar Kyoto (WKY) rats to SHR led to higher sympathetic activity, independent of renin levels [71,72]. Taken together, this suggests that gut microbiota can stimulate sympathetic drive, possibly by direct ENS–brain interactions or by promoting neuroinflammation. This increased sympathetic activity can contribute to hypertension development directly or indirectly, by stimulating low-grade systemic inflammation.

## 3. Gut Microbiota in Atherosclerosis

### 3.1. Atherosclerosis and Gut Microbiota

Atherosclerosis is a multifactorial process, with lipid metabolism, inflammation, vascular ageing, and blood pressure as key players. Atherosclerosis is closely related to arterial stiffness, which is caused by a loss of elastic fibers and thickening of arteriole walls. Arterial stiffness tends to increase with age and results in a less compliant arterial system and higher pulse wave velocity. The resulting increased shear stress has an aggravating effect on the formation of subsequent atherosclerotic plaques [94,95]. In this process, cholesterol accumulation in vessel walls leads to transformation of macrophages to foam cells after phagocytic uptake of lipid particles. Oxidation of lipids results in cholesterol crystallization, inflammasome activation, and production of proinflammatory cytokines such as TNF-alpha and IL-1B. Statins have been proven effective in preventing atherosclerotic events, not only by lowering low-density lipoprotein (LDL) cholesterol, but also through anti-inflammatory effects [96]. The Canakinumab Anti-Inflammatory Thrombosis Outcomes Study (CANTOS)-trial underlined the role of inflammation in atherosclerosis by demonstrating that treatment with canakinumab, a monoclonal inhibitor of IL-1B, lowers the incidence of cardiovascular events [97].

An atherosclerotic plaque was shown to be a microbial environment on itself, containing microbes such as *Streptococcus*, *Pseudomonas*, *Klebsiella*, *Veillonella* spp., and *Chlamydia pneumoniae* [98,99,100]. Most studies could not relate plaque microbiota composition to outcomes such as plaque vulnerability, rupture, or cardiovascular events [101,102]. It was suggested that pathogenic bacteria originating from oral or gut microbiomes make vessel walls more prone to plaque formation, either by direct infection of the vessel wall or by distant infections eliciting an auto-immune inflammatory reaction through molecular mimicry [103,104]. Interventions with antibiotic treatment as secondary prevention, targeted at eliminating plaque microbiota, did not result in lower incidence of cardiovascular events [105,106]. Therefore, these studies did not provide evidence for direct vessel wall infection as a causal factor, although some argue that not all microbes were targeted by the antibiotics used and that interventions were too short [104,107].

In humans, cross-sectional studies showed that higher abundance of the *Collinsella* genus, *Enterobacteriaceae*, *Streptococcaceae*, and *Klebsiella* spp., and lower abundance of SCFA-producing bacteria *Eubacterium*, *Roseburia*, and *Ruminococcaceae* spp. in the gut microbiota of patients with symptomatic atherosclerosis compared to healthy controls [108,109,110]. Pulse wave velocity, a marker of arterial stiffness, was associated with a lower alpha diversity and lower number of SCFA-producing bacteria such as *Ruminococcaceae* spp. in middle aged women in the TwinUK cohort [111]. Hence, the compositional differences in atherosclerosis overlap with findings in hypertensive patients, which is not surprising considering the shared risk factors and pathogenesis. Causal evidence of gut microbiota composition in atherosclerosis is based on fecal microbiota transplantation (FMT) in animal studies. For example, mice transplanted with a more pro-inflammatory gut microbiota composition from Caspase1^-/-^ mice had 29% larger plaque sizes than controls [112]. Alternatively, gut microbiota could have indirect proatherogenic effects, by production of pro-atherogenic metabolites. These metabolites could also very well include the metabolites that are described for hypertension, including SCFAs. For the scope of this review, we chose to focus on the role of trimethylaminoxide (TMAO) and bile acids in atherosclerosis.

### 3.2. Trimethylamine-N-Oxide

The role of trimethylamine (TMA) and TMAO in the development of atherosclerosis is an extensively researched topic. The role of gut microbiota in TMAO production is illustrated by Figure 2. TMA is produced by gut microbes, primarily those from the families *Clostridia* and *Enterobacteriaceae*, in the degradation of nutrients such as carnitine, choline, and lecithin, that can be found in dietary products including meat and eggs [113]. After absorption, TMA is oxidized into trimethylamine-N-oxide (TMAO) by the hepatic enzyme flavin mono-oxygenase (FMO)-3 [114]. Plasma levels of TMAO have both a high within-individual and inter-individual variability, which hampers comparison of studies [115]. In addition, TMAO levels are higher in women, presumably due to different expression of the converting enzyme FMO3 and higher excretion rates in men [114]. TMAO is primarily excreted by the kidneys through both glomerular filtration and tubular secretion, which is a reason for increasing TMAO levels with decreasing renal function [116].

Several mechanisms for the role of TMAO in atherosclerosis have been proposed, including the effects TMAO has on inflammation, cholesterol metabolism, and thrombosis. TMAO was shown to increase the production of pro-inflammatory cytokines such as TNF-alpha and IL-1B, and decrease anti-inflammatory cytokines such as IL-10 [117]. In addition, the hepatic enzyme FMO3 appeared to have a regulating function in lipid metabolism. FMO3 knockdown in mice on a high cholesterol diet lowered intestinal lipid absorption and hepatic cholesterol production and stimulated reverse cholesterol transport, thereby restoring cholesterol balance [118]. Lastly, TMAO was reported to induce platelet hyperreactivity, which can facilitate thrombosis, thus causing atherosclerotic thrombotic events [119].

Administration of TMAO indeed promoted atherosclerosis in several mouse models [120,121]. However, there are also several animal studies that could not confirm this association, or even found a protective effect of TMAO [122,123,124,125]. In humans, higher levels of TMAO have been associated with cardiovascular disease incidence in several prospective studies [123,126,127]. Two meta-analyses concluded that elevated TMAO levels were associated with a higher risk of cardiovascular events and a higher all-cause mortality with relative risks ranging between 55% and 62% [122,128].

Nevertheless, a causal effect of TMAO on atherosclerosis has not yet been proven. An elegant way to assess causality is Mendelian randomization, using genetic variants known to modify the exposure to examine the effect on disease [129]. In this case, the prevalence of cardiovascular disease in individuals with single nucleotide polymorphisms (SNPs) known to cause higher levels of TMAO was compared to individuals without these SNPs [130]. Interestingly, in this study, atherosclerotic cardiovascular disease was not more prevalent in the group with genetically predicted higher TMAO levels. Another way to prove causality is to lower TMAO levels with interventions, such as with TMA lyases that lower TMAO by degrading TMA before oxidization [131]. However, results of human intervention studies have not yet been published.

### 3.3. Bile Acids

Bile acid metabolism is dependent on microbial modifications in the gut (Figure 3) and this interaction was previously shown to affect inflammatory bowel disease and hyperinsulinemia [132,133]. Primary bile acids are synthesized by the liver, which converts hydrophobic cholesterol to hydrophilic primary bile acids [134]. These bile acids are excreted by the gall bladder and reabsorbed in the terminal ileum by sodium-dependent bile acid transporters [135]. Bile acids affect gut microbiota composition and inhibit microbial growth in the small intestines [136]. A small proportion of bile acids reaches the colon, where microbiota convert primary bile acids to secondary bile acids by several modifications, including deconjugation, 7α-dehydroxylation, and 7α-hydrogenation [137]. Secondary bile acids are hydrophobic and therefore easily absorbed by colonocytes and taken up into the systemic circulation. Only an estimated proportion of 5% of bile acids escape the enterohepatic cycle and are excreted [138]. Bile acids also affect diverse metabolic pathways through Takeda G-protein coupled receptor 5 (TGR5) and the nuclear farnesoid X receptor (FXR), both of which have a preference for secondary bile acids. The composition of the microbiota and the microbial community’s enzymatic repertoire determine the secondary bile acid profile [139]. The impact of gut microbiota on the bile acid pool was illustrated by a study showing that germ-free mice had a 71% decreased bile acid pool compared to controls [140]. Interestingly, the bile acid metabolism interacts with the TMAO pathway, as FXR has been shown to regulate FMO3, the hepatic enzyme that converts TMA in TMAO [114].

TGR5 is expressed in a variety of tissues, including liver, gall bladder, intestines, kidneys, pancreas, muscle, and adipose tissue, but can also be found on leukocytes, macrophages, and endothelial cells [141]. A TGR5 agonist (INT-777) was shown to have immunosuppressive effects, including reduced pro-inflammatory cytokine production by macrophages and attenuation of atherosclerotic plaque formation in LDL^−/−^ mice [142,143]. Translation of findings from animal studies on TGR5 to humans in other contexts has not always been successful. Despite beneficial metabolic effects of TGR5 agonists in mice, including lower glucose levels and improved lipid profiles, the TGR5 agonist SB-756050 increased fasting glucose levels compared to placebo in human subjects with type 2 diabetes [144]. TGR5 agonists had limited adverse effects in this trial, which is surprising considering the number of tissues that express this receptor. In animal models, TGR5 agonists have been associated with increased gastrointestinal motility, a potential higher incidence of biliary stones, lower vascular tone and blood pressure, and itching [145].

Atherogenic mice models with FXR knock-out showed conflicting findings, with both increased and decreased atherosclerosis [146,147,148]. However, administration of synthetic FXR agonists to atherogenic mice prevented plaque formation in three studies, presumably by lipid-lowering and anti-inflammatory effects [149,150,151]. Although the FXR agonist obeticholic acid (OCA) lowered hepatic fat in human subjects with non-alcoholic steatohepatitis (NASH), it had paradoxical effects on cholesterol levels, increasing LDL and decreasing high-density lipoprotein (HDL) cholesterol [152].

Dual agents that target both TGR5 and FXR might have more therapeutic potential. Animal studies on the effect of dual agonists reported beneficial effects on metabolic syndrome, NASH, cholangiopathy, progression of diabetic nephropathy, and atherosclerosis [153,154,155,156,157]. In a mouse model for atherosclerosis, dual targeting with INT-767 seemed to be more effective in attenuating atherosclerosis than separate effects on TGR5 and FXR [153]. All in all, although findings in animal studies are promising, it remains to be seen whether these results can be translated to humans, especially considering the substantial differences in atherosclerosis pathogenesis and bile acid metabolism between men and mice.

## 4. Therapeutic Strategies

The changes in gut microbiota composition and gut metabolites discussed in this review could all be potential therapeutic targets in the treatment of atherosclerosis and hypertension. The most direct ways of altering gut microbiota composition are oral supplementation of specific microbial strains and fecal microbiota transplantation (FMT).

Probiotics containing SCFA-producing microbes including *Bifidobacterium*, *Enterococcus*, and *Lactobacillus* were suggested to have a variety of health benefits including anti-inflammatory and beneficial metabolic effects [158]. In addition, oral treatment with specific *Bifidobacterium*, *Lactobacillus*, and SCFA-producing *Anaerobutyricum soehngenii* species had modest blood pressure lowering effects in humans [43,159]. However, our understanding of mechanisms is based on animal research. Evidence in humans is limited and inconclusive due to heterogeneity in investigational products and study designs [160]. Therefore, the effect of specific strains is often unclear, which is one of the reasons that probiotics are marketed as nutritional supplements rather than medication [160].

Probiotic efficacy is both disease-specific and strain-specific [161], underlining the need for well-designed trials that survey gut microbiota composition before and after the intervention. Preferably, this should be measured with metagenomic sequencing (as opposed to 16S rRNA sequencing) in order to provide species-level resolution to compositional data. Another advantage of this technique is the potential to assess differences in gut microbiota functionality, as differences in microbiota composition do not always match differences in function. In addition, the gut microbiome has a spatial dimension, with composition gradients along the different parts of the intestinal tract, yet due to sampling difficulties, fecal samples are used as a proxy for the entire extent of the intestinal tract lumen. Localized sampling would aid in deciphering the actual biology in the intestine.

Alternatively, FMT could be used to optimize microbiota composition in individuals at risk for cardiovascular disease. FMT has been shown to be efficacious with limited adverse effects [162]. However, optimal FMT approaches, including donor selection, screening and preparation, have yet to be defined [163,164]. In addition, the long-term effects of FMT are not clear, since the follow-up in most studies is less than a year. As our understanding of the gut microbiome progresses, so does our knowledge of potential risks of FMT. To illustrate, bacteriophages—long understudied yet now known to play an important role in the microbiome—were shown to be transferred from donor to host by FMT, with uncertain implications [165].

To date, only one FMT trial targeted cardiovascular risk by transplantation from lean vegan donors to meat-consuming subjects with metabolic syndrome in order to lower TMAO levels [166]. Despite alterations in gut microbiota composition, TMAO levels did not change upon this intervention. Other FMT trials in obesity and metabolic syndrome also showed that effects on microbiota composition and glucose metabolism are small and transient, underlining the importance of pre-screening in order to select recipients most likely to respond [167,168]. To that end, a better understanding of the structural and functional aspects of the microbiota that affect hypertension and atherosclerosis incidence is needed.

Prebiotics and dietary interventions target gut microbiota composition indirectly. Prebiotics selectively stimulate specific microbes in the colon. Prebiotics are often fibers, although not all fibers are prebiotics [169]. Prebiotics were shown to stimulate growth of SCFA-producing microbes such as *Bifidobacterium* and *Lactobacillus*. Diet also has a substantial influence on gut microbiota composition. Dietary interventions such as the DASH and the Mediterranean diet were shown to lower cardiovascular risk [170,171]. However, since dietary interventions are multifaceted, it is difficult to point out what mechanisms explain the beneficial effects.

In summary, multiple interventions could target gut microbiota composition and its associated metabolites, ranging from targeted approaches to more accessible but non-specific interventions. However, translation of findings from animal studies to humans is needed, preferably by prospective cohort studies using metagenomic sequencing that can also assess microbiome functionality. In addition, adjusting for confounders when assessing associations between microbiota and cardiovascular disease is vital, since microbiota composition is shaped by a combination of lifestyle factors, health conditions, and medication use.

## 5. Conclusions

The pathways by which gut microbiota affect hypertension and atherosclerosis are diverse and often interact, as shown in Figure 4. Gut microbiota produce or convert metabolites, produce substrates needed for production of metabolites elsewhere, and are involved in regulating local intestinal homeostasis, resulting in a wide range of potential therapeutic targets. However, our understanding of mechanisms is mainly based on animal research and translation to humans remains challenging, as illustrated by developments in bile acid receptors research. Longitudinal studies in human subjects are needed to identify beneficial or adverse characteristics of gut microbiota structure and functionality, in order to better target potential therapeutic strategies.

## Figures and Tables

**Figure 1 nutrients-12-02982-f001:**
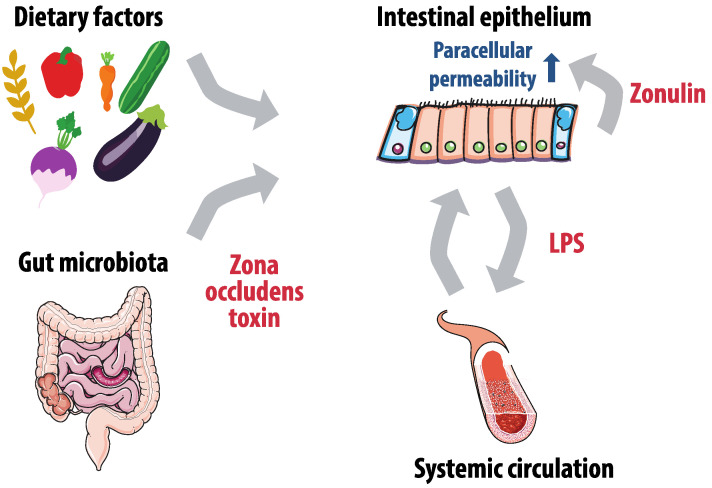
Gut microbiota, gut permeability and lipopolysaccharides (LPS) absorption. Paracellular permeability of the intestinal epithelium is affected by zonulin production of the basal lamina, dietary factors and gut microbiota that produce zone occludens toxin. Increased permeability leads to more LPS translocation to the systemic circulation, which has a pro-inflammatory effect and further increases gut permeability.

**Figure 2 nutrients-12-02982-f002:**
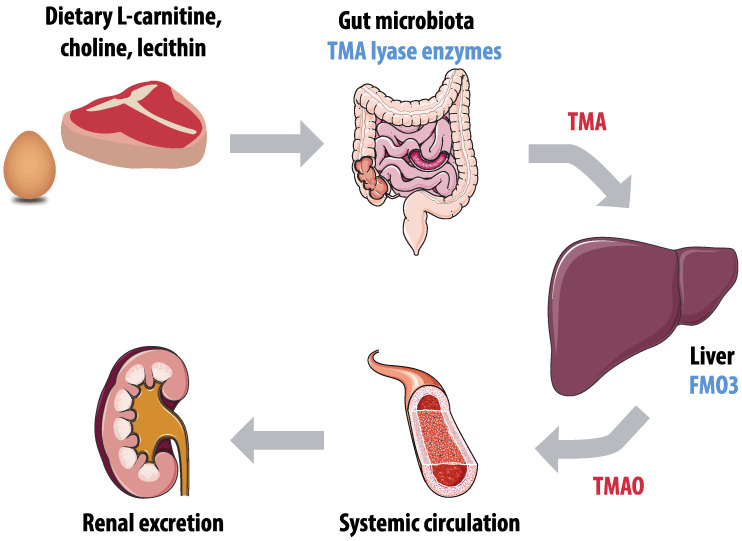
Production of trimethylamine-N-oxide (TMAO). Gut microbiota enzymes, including trimethylamine (TMA) lyase, convert dietary L-carnitine, choline, and lecithin into TMA. The hepatic enzyme flavin mono-oxygenase 3 (FMO3) converts TMA into TMAO, and TMAO is primarily excreted by the kidneys.

**Figure 3 nutrients-12-02982-f003:**
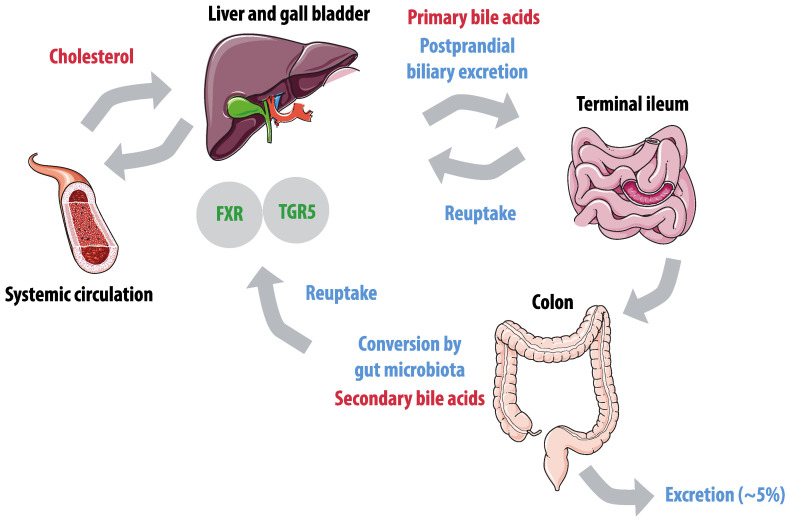
Enterohepatic cycle of bile acids. Hepatic conversion of cholesterol results in primary bile acids, that are excreted postprandially by the gallbladder. Active reuptake takes place in the terminal ileum. In the colon, primary bile acids are converted to secondary bile acids by gut microbiota, and passively reabsorbed. Farnesoid X receptor (FXR) and Takeda G-protein coupled receptor 5 (TGR5) have a preference for secondary bile acids.

**Figure 4 nutrients-12-02982-f004:**
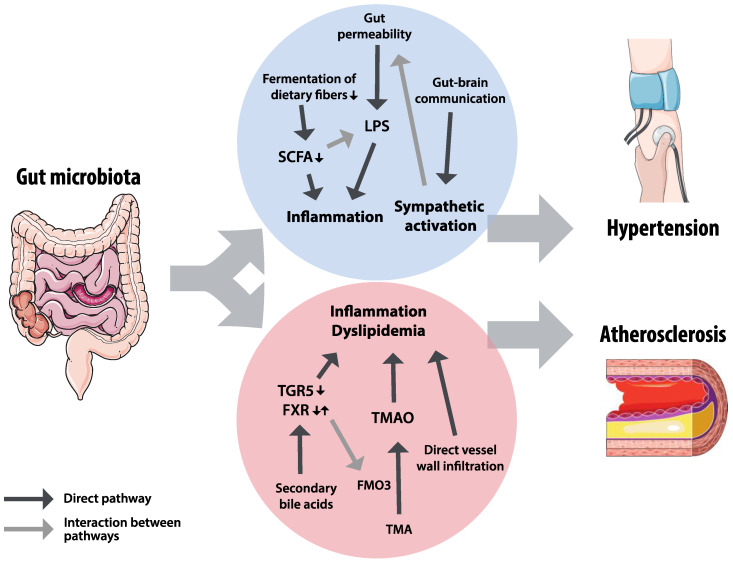
Summary of hypothesized pathways for the effects of gut microbiota on hypertension and atherosclerosis. Gut microbiota could affect hypertension through inflammatory factors, influenced by short chain fatty acids (SCFAs) and lipopolysaccharides (LPS), and through sympathetic activation by gut–brain interactions. The effects on inflammation and dyslipidemia in atherosclerosis could be mediated by bile acid receptors Takeda G-protein-coupled receptor 5 (TGR5) and farnesoid X receptor (FXR), trimethylamine-N-oxide (TMAO) and trimethylamine (TMA), and direct vessel infiltration of microbiota. The grey arrows indicate interactions between pathways: FXR regulates the TMAO-converting enzyme flavin mono-oxygenase 3 (FMO3), sympathetic activation increases gut permeability, and short chain fatty acids can attenuate the inflammatory effects of LPS.

**Table 1 nutrients-12-02982-t001:** Cross-sectional studies on gut microbiota composition in hypertension in humans.

Author	Population	Hypertension Definition	Sequencing Method	Higher Abundance in HT or Higher BP	Lower Abundance in HT or Higher BP	Alpha Diversity in HT or Higher BP	Covariates in Analyses	Ref.
Dan et al. 2019	67 HT, 62 controls	SBP ≥ 140 or DBP ≥ 90 mmHg	16S	Acetobacteroides, Alistipes, Bacteroides, Christensenella, Clostridium sensu stricto, Desulfovibrio, Parabacteroides *	Acetobacteroides, Clostridium, Coprobacter, Enterococcus, Enterorhabdus, Lachnospiracea, Lactobacillus, Paraprevotella, Prevotella, Romboutsia, Ruminococcus, Veillonella *	No difference	Unadjusted	[29]
De la Cuesta-Zuluaga et al. 2019	441 subjects	No hypertension groups	16S	NR	NR	Lower	Unadjusted	[30]
Huart et al. 2019	38 HT, 7 pre-HT, 9 controls	Antihypertensive medication use, mean 24 h BP SBP ≥ 130 or DBP ≥ 80 mmHg	16S	Clostridum sensu stricto	Ruminococcaceae, Clostridiales	NR	Unadjusted	[31]
Jackson et al. 2019	756 HT, 1790 controls	Self-report or antihypertensive medication use	16S	Lactobacillaceae, Streptococcaceae	Dehalobacteriaceae, Christensenellaceae, Oxalobacteraceae, Mollicutes, Rikenellaceae, Clostridia, Anaeroplasmataceae, Peptococcaceae	Lower	Age	[32]
Kim et al. 2018	22 HT, 18 controls	SBP ≥ 140 mmHg	Shotgun	Parabacteroides johnsonii, Eubacterium siraeum, Alistipes finegoldii	Bacteroides thetaiotaomicron	NR	Unadjusted	[33]
Li et al. 2017	99 HT, 56 pre-HT, 41 controls	SBP ≥ 140 or DBP ≥ 90 mmHg	Shotgun	Prevotella, Klebsiella, Desulfovibrio	Faecalibacterium, Oscillibacter, Roseburia, Bifidobacterium, Coprococcus, Butyrivibrio	Lower	Unadjusted	[34]
Sun et al. 2019	529 subjects (183 HT)	Antihypertensive medication use or elevated office BP: SBP ≥ 140 or DBP ≥ 90 mmHg	16S	Anaerovorax, Butyricicoccus, Cellulosibacter, Clostridium IV, Methanobrevibacter, Mogibacterium, Oscillibacter, Oxalobacter, Papillobacter, Sporobacter, Vampirovibrio	Anaeroglobus, Atopobium, Lactobacillus, Megaspheara, Pseudocitrobacter, Rothia,	Lower	Age, ethnicity, sex, study center, sequencing run, education, smoking, physical activity, diet quality score	[35]
Verhaar et al. 2020	4672 subjects	No hypertension groups		Streptococcus	Roseburia, Clostridium sensu stricto, Roseburia hominis, Romboutsia, Ruminococcaceae, Enterorhabdus	Lower	Age, sex, BMI, smoking status, antihypertensive medication, diabetes	[36]
Yan et al. 2017	60 HT, 60 controls	SBP ≥ 140 or DBP ≥ 90 mmHg	Shotgun	Klebsiella, Streptococcus, Parabacteroides	Roseburia, Faecalibacterium prausnitzii	Lower	Not adjusted, but age, sex−, and BMI-matched	[37]
Yang et al. 2015	7 HT, 10 controls	SBP ≥ 125 mmHg	16S	NR	NR	Lower	Unadjusted	[27]

BP = blood pressure, DBP = diastolic blood pressure, SBP = systolic blood pressure, HT = hypertensive, NR = not reported, * = selection of the microbiota listed by this paper.

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
