# Peer review of "Gut Microbiota in Hypertension and Atherosclerosis: A Review"

_nutrients, 2020, doi:10.3390/nu12102982_

Round 1

Reviewer 1 Report

This review manuscript from Verhaar et al summarized relationship among gut microflora and hypertension and atherosclerosis. The cover topics are very wide. If authors intend this review for a broad summary, this review does its jobs. I have a few comments to improve this manuscript.

After broad introduction, authors stated gut microflora and hypertension. Microflora composition was summarized well in Table 1. However, short cain fatty acids and Gut permeability/LPS do not have associated figure/table. Readers will appreciate visual aids for these sections. Please expand Figure 1 or make independent figures.

I was not fully understood why gut-brain interactions are included to this review article. I see the interaction of brain for this topics but how about other organs? If other organs do not interact with hypertension, or Brain is a major organ to interact with hyperension, please clarify this in the section.

Then, authors summarized atherosclerosis with gut microbiota.

The interactions with atherosclerosis and microbiota associated chemicals were explained in this section. I could not figure out how gut micobiota interact with secondary bile acids and TMAO in the figure 1. Please add extra figures to support contents visually.

Finally, therapeutic strategies were briefly mentioned. This section should be 4 not 3. Probiotics and fecal transplantation were suggested. Authors introduced current situations well. I have no issue for this section.

And Conclusion should be 5 not 4.

Figure 1. It was unclear the meaning of arrow color (black and grey). Please clarify. It was unclear how gut microbita (alteration) directly/indirectly associates with arrows. Please clarify this too.

Author Response

We thank the reviewer for the constructive remarks and suggestions.

We understand that visual illustration of the discussed pathways would be helpful. Therefore, we made additional figures that illustrate how gut microbiota affect the production of the discussed metabolites (Figure 1-3). For the short chain fatty acids, we now describe in more detail how production of these metabolites is affected by gut microbiota (line 110-115):

“Fecal and plasma levels of SCFA are associated with the abundance of SCFA-producing microbiota in the gut and the intake of dietary fibers [50–52]. Butyrate-producing microbiota include bacteria from the families Ruminococcaceae and Lachnospiraceae, but also bacteria such as Anaerobutyricum hallii and Anaerostipes spp. Acetate and propionate are mainly produced by Bifidobacterium spp. and mucin-degrading bacteria such as Akkermansia muciniphila [53].” 

The summary figure (Figure 4) aims to summarize the effects of the different pathways that are discussed in this review and therefore lacks mechanistic detail with regard to the influence of gut microbiota on the production of gut metabolites. However, we hope that the extra figures (Figure 1-3) in the revised version of the manuscript are helpful in illustrating these mechanisms. We added a legend to Figure 1 to explain the colors of the arrows more clearly.

The review’s structure is based on potential pathways, and not organ systems per se. For instance, we describe the effects of short chain fatty acids on vasculature and kidneys, but this is embedded in the short chain fatty acid section, since this is the associated pathway. Since the title of the section on gut-brain interactions might give an organ-centered impression, we changed the title of this section to ‘Gut-brain interactions and sympathetic activation’. We included gut-brain interactions in this review because sympathetic activation is one of the key factors in hypertension pathogenesis. The enteric nervous system is sometimes described as the ‘second brain’ and is a complex system that consists of approximately 500 million neurons. The available literature points to a role for the ENS in sympathetic activation. Therefore, we feel that discussing the interaction between the enteric nervous system, central nervous system and sympathetic activation is essential in a review on gut microbiota and hypertension. In the revised manuscript, we further underline the importance of the ENS:

“The ENS is a complex system that is sometimes referred to as the ‘second brain’, because of the structural and functional similarities [93].” (line 209-210)

Reviewer 2 Report

This is a comprehensive and well-written review on gut microbiota in hypertension and atherosclerosis.

The authors first give a concise overview of the microbial produced metabolites such as SCFA and LPS in hypertension followed by a concise overview of the microbial produced metabolites such as TMAO and bile acids in atherosclerosis.

I have some minor suggestions that I think would be important to address:

  • Line 105: “These differences point to a role for SCFAs and LPS in blood pressure regulation.”
    The authors summarize the changes in SCFA, LPS producing bacteria. However, the authors should provide evidence from the literature on associations between SCFA, LPS and blood pressure differences.
  • Section 3.3: This section nicely summarizes known roles of bile acids in driving of pro-atherogenic changes; however, it fails to make a direct connection to the gut microbiota. Please address this concern and provide more evidence of how changes in the gut microbiota affect the bile acid composition and metabolism.
  • Section 3. Therapeutic strategies: Please include more details on current drawbacks of using FMT and future areas of research that will improve the safety and efficacy of this strategy.
  • Conclusion: Could the authors comment on differences in microbial composition detected from 16S sequencing of fecal matter vs sequencing of luminal and mucosal communities from the different segments of the intestine? It will be important to include such information since the mentioned metabolites get produced/metabolized in different segments along the length of the intestine and sequencing the fecal microbiota does not necessarily reflect the different microbial communities that colonize different segments of the intestine.

Author Response

We thank the reviewer for the kind words and constructive suggestions that helped us to improve the manuscript. Please find our itemized response to the suggestions below.

  • We agree that the final sentence of section 2.1 is too suggestive as a conclusion to this section, since underlying mechanisms are not discussed until the following sections. We have amended the sentence as follows:

“These differences point to a role for SCFAs and LPS in hypertension, although the direction of this association is unclear.” (line 104-105)

  • In the revised manuscript, we underline the role of gut microbiota in the production of secondary bile acids. In brief, TGR5 and FXR have a preference for secondary bile acids, and the production of these bile acids is dependent on modifications by gut microbiota.

“A small proportion of bile acids reaches the colon, where microbiota convert primary bile acids to secondary bile acids by several modifications, including deconjugation, 7α-dehydroxylation and 7α-hydrogenation [142].” (line 311-313)

“The composition of the microbiota and the microbial community’s enzymatic repertoire determine the secondary bile acid profile [144]. The impact of gut microbiota on the bile acid pool was illustrated by a study showing that germ-free mice had a 71% decreased bile acid pool compared to controls [145].” (line 318-321)

  • We discuss FMT now more elaborately in the section on therapeutic strategies, including the considerations mentioned by the reviewer:

“However, optimal FMT approaches, including donor selection, screening and preparation, have yet to be defined [168,169]. In addition, the long term effects of FMT are not clear, since the follow-up in most studies is less than a year.  As our understanding of the gut microbiome progresses, so does our knowledge of potential risks of FMT. To illustrate, bacteriophages – long understudied yet now known to play an important role in the microbiome - were shown to be transferred from donor to host by FMT, with uncertain implications [170].” (line 374-379)

  • Indeed, spatial dimension of microbial composition is not reflected by the sequencing results of fecal samples. We added this consideration to the revised manuscript:

“In addition, the gut microbiome has a spatial dimension, with composition gradients along the different parts of the intestinal tract, yet due to sampling difficulties, fecal samples are used as a proxy for the entire extent of the intestinal tract lumen. Localized sampling would aid in deciphering the actual biology in the intestine.” (line 368-371)